# Effect of Alkali on the Microbial Community and Aroma Profile of Chinese Steamed Bread Prepared with Chinese Traditional Starter

**DOI:** 10.3390/foods12030617

**Published:** 2023-02-01

**Authors:** Ning Tang, Xiaolong Xing, Huipin Li, Honggang Jiao, Shengxin Ji, Zhilu Ai

**Affiliations:** 1College of Food Science and Technology, Henan Agricultural University, 63 Nongye Rd., Zhengzhou 450002, China; 2College of Biology and Food, Shangqiu Normal University, Shangqiu 476000, China; 3National R&D Center for Frozen Rice&Wheat Products Processing Technology, Zhengzhou 450002, China; 4Key Laboratory of Staple Grain Processing, Ministry of Agriculture and Rural Affairs, Zhengzhou 450002, China

**Keywords:** microbial composition, aroma compounds, traditional fermented wheat flour food, alkali addition, high-throughput sequencing, synthetic microbial community

## Abstract

Alkali is an indispensable additive in Chinese steamed bread (CSB) production. This work aimed to evaluate the key roles of alkali in the microbial community of dough fermented using Chinese traditional starter (CTS) and the aroma profiles of CSB. The dominant fungi in CTS and fermented dough were members of the phylum Ascomycota and the genus *Saccharomyces*. *Pediococcus*, *Companilactobacillus*, and *Weissella* were the dominant bacterial genera in CTS and fermented dough. Adding alkali could retain the types of dominant yeasts and LAB derived from CTS, decrease the relative abundance of *Companilactobacillus crustorum* and *Weissella cibaria*, and increase that of *Pediococcus pentosaceus*, in fermented dough. Principal component analysis (PCA) indicated that adding alkali decreased the content of sourness-related volatiles in CSB fermented by CTS. Correlation analysis showed that *Pediococcus* and *Weissella* in fermented dough were positively correlated with the lipid oxidation flavor-related compounds in CSB, and *Lactobacillus* was positively correlated with sourness-related aroma compounds. Synthetic microbial community experiments indicated that CSB fermented by the starter containing *P. pentosaceus* possessed a strong aroma, and adding alkali weakened the flavor intensity. Alkali addition could promote the formation of ethyl acetate and methyl acetate with a pleasant fruity aroma in *W. cibaria*-associated CSB.

## 1. Introduction

Chinese steamed bread (CSB) is a fermented wheat flour food, a traditional staple food in northern China with >1500 years of history [1]. As an ancient biotechnological product, Chinese traditional starter (CTS) is a kind of starter culture for preparing CSB. CTS is made by using cereal flour, water, and various materials of regional origin (e.g., juice, fruit, and wine) according to local dietary customs [2]. Generally, because of the complex materials and different processing methods used, CTS contains more abundant microbes than modern commercial yeast starters [3]. Therefore, traditional CSB is considered richer in nutrients and to have a more attractive smell than commercial yeast-based CSB [4].

Accurate temperature, suitable acidity, and appropriate moisture are basic requirements for dough fermentation. Thus, CSB is usually made by experienced homemakers or workers. Historically, CSB makers used vegetation greywater or trona to neutralize the acidity produced during dough fermentation, thereby decreasing the unpleasant sourness of CSB. Because the quality of these ancient neutralizers was uncertain, the quality of the CSB product was also unstable. Fortunately, the advent of industrial food-grade alkali, that is, Na_2_CO_3_, provided a reliable solution [5]. It rapidly replaced the ancient neutralizers and is widely used in cereal foods on a global scale, such as in wheat noodles [6], corn tortillas [7], and pretzels [8]. Alkali has even become a primary additive in the manufacture of CSB. Some recent reports have revealed that adding alkali could improve the specific volume and texture properties with increased crumb hardness and resilience of buckwheat CSB [9], and Na_2_CO_3_ could improve the flavor of CSB by inhibiting aroma-negative compounds [10].

CTSs are complex and stable biological ecosystems, containing bacteria, yeasts, and molds. The microbial composition and main strains have been the subject of many studies. The yeasts possess superior gas generation capacity and make CSB spongy and fluffy. The reported relevant yeast species include *Saccharomyces cerevisiae*, *Kazachstania humilis*, *Pichia kudriavzevii*, and so on [3,11]. The lactic acid bacteria (LAB) in CTSs include *Fructilactobacillus sanfranciscensis*, *Pediococcus pentosaceus*, *Companilactobacillus crustorum*, *Weissella cibaria*, *Lactiplantibacillus plantarum*, etc. [2,12]. CTS-derived yeasts and LAB are considered to have a vital role in improving the flavor of wheat flour products because of their high metabolic activity in producing flavor compounds and their precursors [13]. Suo et al. [4] reported that *Pediococcus* was the predominant genus in CTS by high-throughput sequencing, and there were far more flavor compounds in CSB compared with commercial yeast-based CSB. Hu et al. [14] used a synthetic microbial community technique that was the combination of LAB with yeast strains derived from sourdoughs as new starter cultures for wheat flour dough fermentation, and found that the doughs inoculated with *L. plantarum* and *S. cerevisiae* cultures could replace naturally fermented doughs and improve the specific volume, texture, and aroma of bread. The combination of high-throughput sequencing and synthetic microbial community techniques could provide convenience for the study on the relationship between the microorganisms in CTSs and the aroma profiles of CSBs.

However, systematic study of the effect of adding alkali on the microbial composition in dough fermented by CTS and on the aroma profile of CSB has been ignored. Therefore, we investigated CTSs sampled from different regions in northern China, analyzed the fungal and bacterial communities in CTSs and fermented doughs with alkali addition, and then detected the aroma compounds in the corresponding CSBs by gas chromatography-mass spectrometry (GC-MS). Finally, we prepared synthetic microbial community starters for dough fermentation using LAB combined with *S. cerevisiae* cultures and determined the content of aroma compounds of CSBs fermented by these starters. This study bridges gaps in current knowledge and provides novel insights into the mechanisms underlying the improvement of CSB flavor via alkali addition.

## 2. Materials and Methods

### 2.1. Materials

Hand-made or workshop-manufactured CTSs were sampled from eight cities in northern China (Table 1): Wuwei City, Gansu (WW); Weinan City, Shaanxi (WN); Yuncheng City, Shanxi (YC); Luoyang City, Henan (LY); Lankao City, Henan (LK); Heze City, Shandong (HZ); Binzhou City, Shandong (BZ); and Weihai City, Shandong (WH). All-purpose wheat flour (produced on 16 July 2021) was purchased from JinYuan Kite Flour Co., Ltd., Zhengzhou, China. Alkali (food-grade Na_2_CO_3_; Bakerdream Co., Ltd., Weifang, China) was purchased from a local supermarket.

### 2.2. Manufacture of CSB using CTS

According to the recommendations of each CTS maker, the optimized method of CTS dough fermentation was as follows: First, 15 g CTS and 0.45 g alkali were pre-dissolved in 100 g aseptic water at 30 °C for 10 min, added into 185 g wheat flour, and mixed in a dough mixing machine (Model AB-DCN03, Zhuhai Appliance Co., Ltd., Zhuhai, China) for 20 min. Then, 60 g of each dough was sampled to make a well-proportioned hemispherical shape and put into a fermentation cabinet to proof at 35 °C and 80% relative humidity. The fermentation time referenced the method of Suo et al. [4]. When the proofing process was completed, the volume of the dough had doubled, and the inside was a dense honeycomb structure, with slightly sour flavor. Alkali-added fermented dough (AFD) was sampled at the end of the proofing process, with alkali-free fermented dough (FD) as control. Finally, the fermented dough was steamed above boiling water in a pot for 25 min, then left to rest at room temperature for 10 min to obtain CSB sample.

### 2.3. Isolation and Identification of Dominant Yeast and LAB Strain from CTS

Yeast and LAB from CTS were isolated and cultivated using Yeast Peptone Dextrose (YPD) and De-Man Rogosa Sharpe (MRS) medium (Beijing AoBoX Bio-Tech Co., Ltd., Beijing, China), respectively. The strains of yeast and LAB were identified using homology alignment of conserved DNA sequences. DNA was extracted from single colonies using a fungal or bacterial DNA extraction kit (Omega Inc., Norcross, GA, USA). PCR was used to amplify the internal transcribed spacer (ITS) rRNA gene of yeast and the 16S rRNA gene of LAB. The universal primer pairs used were ITS1/ITS4 and 27F/1492R, respectively. The program of amplification referred to the Kit instructions (Vazyme Biotech Co., Ltd., Nanjing, China). The conditions were denaturation at 95 °C for 3 min; 35 cycles of 95 °C for 15 s, 58 °C for 30 s, elongation at 72 °C for 90 s; and a final extension at 72 °C for 5 min. The total reaction volume was a 20 μL mixture containing 10 μL of 2 × Phanta Max Buffer, 0.4 μL of 10 mM dNTP Mix, 0.8 μL of forward primer (10 μM) and reverse primer (10 μM), 0.4 μL of Phanta Max Super-Fidelity DNA Polymerase, and 1 μL of template DNA. The PCR products were sequenced by Wuhan AuGCT Biotechnology Co., Ltd. The sequencing data were submitted to GenBank (https://www.ncbi.nlm.nih.gov/genbank/, (accessed on 27 June 2022)). Strains were stored in 30% glycerin solution at −80 °C.

### 2.4. Manufacture of CSB Fermented Using Cultivated Yeast and LAB

The yeast and LAB strains were cultivated in the corresponding liquid medium until the beginning of the stationary growth phase. Cells were collected from these cultures by centrifugation (7000 rpm, 5 min) at 20 °C and washed twice with aseptic distilled water. Collected cells were used immediately to make CSBs. CSBs fermented by yeast and LAB were manufactured as follows: Firstly, collected cells were directly added into 100 g aseptic distilled water and 200 g wheat flour (the added concentrations of yeast and LAB cells were 5 × 10^8^ and 1 × 10^9^ CFU/g in the unfermented doughs, respectively), and they were mixed in a dough mixing machine for 20 min. Then, 60 g of each dough was sampled to make a well-proportioned hemispherical shape and put into a fermentation cabinet to proof at 35 °C and 80% relative humidity. When the proofing process was completed, the volume of the dough had doubled, and the inside was a dense honeycomb structure, with slightly sour flavor. Finally, the fermented dough was steamed above boiling water in a pot for 25 min, then left to rest at room temperature for 10 min to obtain CSB sample. Details of the strains are given in Appendix A.

### 2.5. Specific Volume, pH, and Total Titratable Acidity (TTA) of Dough Fermented by CTS

Unfermented dough (20 g) was put into a graduated cylinder. The proofing conditions were the same as those used for preparation of CSB. The final volume was recorded. The specific volume was the ratio of the final volume to the initial weight.

The pH and TTA values were measured according to the previously described method of Yan et al. [15]. Ten grams of each fermented dough and 90 mL aseptic water were put into a sterile sampling bag and homogenized using a homogenizer (Model LC-11L, Shanghai JingXin Co., Ltd., Shanghai, China) until the mixture was thoroughly suspended. The pH of fermented dough was determined using a glass electrode pH meter (Model H01-G, Shanghai Mettler Toledo Co., Ltd., Shanghai, China). The TTA was determined by titration against 0.1 M NaOH solution to pH 8.5 and denoted by the consumed volume of NaOH solution.

### 2.6. High-Throughput Sequencing

Total DNA was extracted from fermented dough using a total DNA extraction kit (Omega Inc.). The hypervariable regions of fungi 18S rRNA gene were amplified with primers SSU0817F (5′-TTAGCATGGAATAATRRAATAGGA-3′) and 1196R (5′-TCTGGACCTGGTGAGTTTCC-3′). For bacteria, the V3 and V4 hypervariable regions of 16S rRNA were amplified using primers 338F (5′-ACTCCTACGGGAGGCAGCAG-3′) and 806R (5′-GGACTACHVGGGTWTCTAAT-3′). Purified amplicons were pooled in equimolar amounts, and paired-end sequencing was performed on Illumina MiSeq PE300 platform/NovaSeq PE250 platform according to standard protocols. Operational taxonomic units (OTUs) with 97% similarity cutoff were clustered using UPARSE version 7.1.

### 2.7. Detection of Volatile Compounds in CSB

According to the GC-MS method of Xi et al. [10], the volatile compounds of CSBs were extracted using 50/30 μm DVB/CAR/PDMS (divinylbenzene/carboxen/polydimethylsiloxane)-coated fibers (Thermo Scientific, Wilmington, DE, USA) by solid-phase microextraction (SPME) and separated on an analytical TG-WaxMS silica capillary column (30 m × 0.25 mm× 0.25 μm, Thermo Scientific). Ultra-high-purity helium was used as the carrier gas at a constant flow rate of 1.0 mL/min. The GC oven temperature was programmed to increase from 40 to 100 °C at 6 °C/min, held at 100 °C for 3 min, increased to 230 °C at 10 °C/min, and then held for 6 min. Electron-impact mass spectra were generated at 70 eV. The ion source and transfer line temperatures were set at 240 °C. The full-scan mode was used to detect all of the compounds in the mass-to-charge ratio range of 33–400 *m/z*.

Volatile compounds were identified individually by matching mass spectra with the NIST20 library database (National Institute of Standards and Technology, Gaithersburg, MD, USA). Ethyl hexanoate (10 μL, 10 mg/L) was added into samples as an internal standard for semi-quantitative calculation.

### 2.8. Statistical Analysis

GraphPad Prism 8 (GraphPad Software Inc., San Diego, CA, USA) software was used for statistical analysis, with differences being considered statistically significant at *p* < 0.05.

## 3. Results and Discussion

### 3.1. Specific Volume, pH, and TTA of Fermented Dough

The leavening capacity of starter is the most important factor that affects the quality of steamed bread. The specific volumes, pH, and TTA values of fermented doughs were determined and are shown in Figure 1. The specific volumes of the fermented doughs were all >2.0 mL/g, and fermented doughs were considered to meet the requirements for making steamed bread [4]. The specific volumes of AFDs were in the range of 2.19–2.37 mL/g, and those of FDs fermented by the same CTSs were 2.06–2.25 mL/g. This was consistent with the previous study of Guo et al. [9], who demonstrated that the addition of 0.1% and 0.2% Na_2_CO_3_ could significantly increase the specific volume of buckwheat CSB to 2.07 and 2.16 mL/g, respectively, by neutralizing the organic acids.

The pH values of AFDs ranged from 4.47 to 6.22, and the TTA values were from 1.6 to 3.3. For FDs, the pH values ranged from 4.16 to 5.44, and the TTA values were from 2.0 to 5.6. The pH and TTA values differed among the doughs fermented by CTSs from different regions. The acidity of fermented dough was mainly related to the microbiota in the starters [16]. The FD from BZ showed the highest pH value (5.44) and the lowest TTA value (2.0) among all of the FD samples, indicating its lower content of organic acids, which could be related to low acid-producing capacity of the microorganisms in the dough ecosystem. Comparing AFDs with the corresponding FDs, adding alkali raised the pH values by 16.9%; the result was similar to that reported by Xi et al. [10].

Overall, the specific volume and acidity of fermented doughs varied in different regions. The addition of alkali could increase specific volume and decrease acidity in fermented dough. These findings can be explained in two ways. First, the complex origin of raw materials and the various production processes of CTSs bring different microorganisms into CTSs, leading to different fermentation properties of doughs, including gas production and acid accumulation [11]. Second, organic acids accumulated in doughs are neutralized by hydroxide from Na_2_CO_3_ dissolution, which promotes the generation of CO_2_, and provides a suitable environment for microorganisms to maintain high gas-producing capacities [9,17].

### 3.2. Fungal Community Analysis in CTS and Fermented Dough

High-throughput sequencing of 18S rRNA gene was performed to analyze the fungal communities in CTSs and fermented doughs. Sequencing quality, community diversity, and community richness were evaluated by the rarefaction curve, the Shannon index, and the Chao1 index, respectively. The sequencing and diversity results of fungi are shown in Appendix A and Figure 2A,B. The number of sequences for each sample was in the range of 35,555–64,086, and the rarefaction curves tended toward horizontal lines, indicating that the sequencing depth was sufficient. There were no significant differences among the fungal Shannon indexes of the CTS, AFD, and FD groups. The fungal Chao1 index of the FD group was significantly higher than that of the CTS group. The results indicated that dough fermentation did not alter the diversity but promoted the richness of the fungal community.

As shown in Figure 3A, five genera belonging to the phylum Ascomycota (Appendix A) were identified in CTSs and fermented doughs, including *Saccharomyces*, *Aspergillus*, *Hyphopichia*, *Fusarium*, and *Pichia*. Consistent with previous research, Ascomycota was the predominant phylum in CTSs, *Saccharomyces* was the dominant genus in the CTS group (98.1%), and their relative abundance increased further in the AFD group [3]. The fungal composition of the FD group was similar to that of the AFD group. Five species were identified in CTSs: *S. cerevisiae*, *Aspergillus penicillioides*, *Hyphopichia burtonii*, *Fusarium graminearum*, and *P. kudriavzevii* (Figure 3B). *S. cerevisiae* is the most important fungus that influences the dough fermentation properties, forming large populations in fermented dough and acting as the major CO_2_ contributor to expand the dough in fermented wheat food preparation [18]. *P. kudriavzevii* is a frequently encountered yeast in sourdough ecosystems [19]. *A. penicillioides* was the subdominant species in terms of relative abundance in the LK (7.8%) and YC (6.7%) CTSs, and it was reported to be the predominant fungus in *sufu*, a traditional Chinese appetizer of fermented soybean curd [20]. *H. burtonii* causes “chalk mold” on bread [21], and *F. graminearum* is a pathogenic fungus that causes wheat scabs [22]; even though they may lead to the deterioration of steamed bread quality or the accumulation of a variety of mycotoxins in steamed bread, they almost disappeared in the mature doughs fermented by CTSs.

### 3.3. Bacterial Community Analysis in CTS and Fermented Dough

Bacterial communities were analyzed by high-throughput sequencing of 16S rRNA gene. The sequencing and diversity results for bacterial communities are shown in Appendix A and Figure 2C,D. The number of sequences for each sample was between 34,143 and 55,287, and the rarefaction curves indicated adequate sequencing depth. The Shannon and Chao1 indexes of the CTS group were higher than those of the AFD and FD groups, suggesting that CTS had higher species abundance and richness than fermented dough.

The phylum Firmicutes was dominant (Appendix A), with relative abundance of 79.8% in the CTS group, and higher abundance in the AFD and FD groups. The subdominant phylum was Proteobacteria, with relative abundance of 20.6% in the CTS group, but this phylum almost disappeared from the AFD and FD groups. Nine bacterial genera (Figure 3C,D) were identified in CTSs and fermented doughs, namely *Pediococcus*, *Companilactobacillus*, *Weissella*, *Fructilactobacillus*, *Levilactobacillus*, *Latilactobacillus*, *Acetobacter*, *Limosilactobacillus*, and *Gluconobacter* [23,24]. Among them, *Pediococcus*, *Companilactobacillus*, *Weissella*, *Fructilactobacillus*, *Levilactobacillus*, *Latilactobacillus*, and *Limosilactobacillus* were common LAB and always found in fermented food. The predominant genus of bacteria in CTS group was *Pediococcus*, and its relative abundance was higher in the FD and AFD groups than in the CTS group, revealing that *Pediococcus* was enriched during dough fermentation. Adding alkali increased the relative abundance of *Pediococcus* from 37.8% in the FD group to 41.1% in the AFD group. *Companilactobacillus* was the sub-abundant genus in the CTS group, and adding alkali decreased its relative abundance in the AFD group. The third abundant genus was *Weissella* in the CTS group. Notably, *Acetobacter* and *Gluconobacter* were detected in the CTS group, but almost none of them was observed in the AFD and FD groups. This could be explained by the following reason: *Acetobacter* and *Gluconobacter* were originated from the raw materials (e.g., *Cucumis melo*) of CTSs, and they could not survive in dough. DNA from dead cells of *Acetobacter* and *Gluconobacter* in CTSs were detected by high-throughput sequencing. In the fermentation process, addition of wheat flour and water led this DNA to greatly diluting, and DNA was also degraded during the leavening process [12].

A Venn plot was used to analyze the similarity and specificity of bacterial species among the CTS, FD, and AFD groups (Figure 4A), indicating that the microorganisms in fermented doughs were almost all derived from CTSs and that adding alkali caused changes between the FD and AFD groups in the types of species originated from CTSs. Co-occurrence network analysis (Figure 4B) of bacterial communities in all CTSs and fermented doughs reflected the dominant species types and coexistence relationships among these samples. The dominant species in CTS, AFD, and FD samples from a given region were the same, indicating alkali addition did not change the types of dominant bacteria derived from CTSs. Ten species (Figure 3E,F) were identified in CTSs and fermented doughs: *P. pentosaceus*, *C. crustorum*, *Pediococcus acidilactici*, *W. cibaria*, *F. sanfranciscensis*, *Levilactobacillus brevis*, *Latilactobacillus curvatus*, *Limosilactobacillus pontis*, *Gluconobacter oxydans*, and *Gluconobacter frateurii*. *P. pentosaceus* and *C. crustorum* are usually found in fermented wheat flour products and demonstrated to be homofermentative LAB in sourdough, contributing significantly to the development of steamed bread flavor [25]. For example, Plessas et al. [26] clarified that *P. pentosaceus* was an efficient starter that improved bread quality and flavor by forming specific organic acids and volatile compounds. *P. pentosaceus* was present in the samples from LY, LK, HZ, and WH and predominated in LK and WH samples. *C. crustorum* was the most abundant species in WN samples; its relative abundance was decreased by adding alkali in fermented doughs. Comasio et al. [27] reported that *C. crustorum* could convert the citrate into L-lactic acid, acetoin, and diacetyl in sourdough and endow wheat bread with a buttery odor. In addition, *W. cibaria* and *F. sanfranciscensis*, which are also obligate heterofermentative LAB [25], dominated in samples from BZ and YC, respectively. The relative abundance of *W. cibaria* increased from 62.8% in CTS to 98.3% in FD from BZ, indicating that *W. cibaria* was highly competitive in this dough ecosystem. *F. sanfranciscensis* was a typical LAB in European type Ⅰ sourdough and the predominant species in CTS, AFD, and FD from YC, with relative abundances of >96.5%. *L. pontis* showed poor competitiveness in dough fermentation, with relative abundance in LK samples decreasing from 25.9% in CTS to 1.2% in FD.

Overall, LAB predominated in the bacterial community of the CTS group because of their high adaptability to the environment of dough fermentation [28]. With the addition of alkali, the relative abundances of *P. pentosaceus*, *P. acidilactici*, *F. sanfranciscensis*, and *L. brevis* increased, while that of *C. crustorum*, *W. cibaria*, and *L. curvatus* declined in the AFD group compared with the FD group (Figure 3E). Adding alkali changed the relative abundance of dominant LAB in fermented dough, which could be explained by the fact that exogenous alkali altered their competitiveness during dough fermentation.

### 3.4. Analysis of Volatile Compounds in CSB

Flavor is a vital factor affecting the sensory properties and acceptance of steamed bread [3]. To explore the flavor characteristics of CSB, the volatile compounds in the steamed breads were detected by GC-MS. Thirty-seven kinds of aroma compounds were identified (Appendix A), including 10 alcohols, 10 aldehydes, eight acids, six ketones, two esters, and one furan, and they were described as specific odors referring to the report of Pétel et al. [29] afterwards.

Alcohols are typical flavor compounds in fermented wheat flour products. Phenylethyl alcohol with flower fragrance [1] and ethanol denoted as alcoholic odor [30] are the most concentrated alcohols in CSB. The phenylethyl alcohol concentration in CSBs from eight regions was in the range of 4.06–40.39 mg/kg, and the highest concentration was detected in alkali-free CSB from WN. The concentration of ethanol in CSBs was between 19.48 mg/kg and 265.38 mg/kg, and its intensity was the highest in alkali-added CSB from BZ. They were reported to be positively correlated with the aroma of wheat bread [31]. That is to say, the higher the concentration of phenylethyl alcohol and ethanol, the higher the acceptance of bread aroma by consumers. In addition, the compound 1-octen-3-ol was demonstrated to give rise to a “mushroom-like” odor [32], which was correlated negatively with bread flavor [31]. Adding alkali obviously decreased the concentration of 1-octen-3-ol concentration from 3.05 to 0.07 mg/kg in CSB from BZ, but its intensity in alkali-free CSB (5.10 mg/kg) increased slightly in comparison with alkali-added CSB (5.37 mg/kg) from HZ. Our results were not absolutely in accordance with the study of Xi et al. [10], who found that adding alkali could improve the flavor of type Ⅰ sourdough CSB by decreasing the concentration of 1-octen-3-ol. That might be related to the more complex composition of LAB in CTS fermented dough than that in type Ⅰ sourdough, and the different sensitivities of LAB to alkali.

The volatile acids are the main source of sourness in traditional CSB, and the primary purpose of adding alkali is to decrease the sour flavor. Butanoic acid (sweaty) was the most concentrated acid, with a concentration of 21.26 mg/kg in CSB from WN, which dropped to 6.07 mg/kg in the corresponding alkali-added CSB. Pentanoic acid (sweaty), hexanoic acid (sweaty), and heptanoic acid (cheese) were identified in all alkali-free CSBs and were decreased or even undetectable in alkali-added CSBs. Similar results were reported by Xi et al. [10], who observed that the addition of alkali could significantly decrease the intensity of butanoic acid in sourdough CSB. The reason could be that adding alkali neutralized the organic acids produced by microbial metabolism in fermented dough.

Aldehydes including hexanal (green), heptanal (fatty), octanal (citrus), (Z)-2-heptenal (green), nonanal (citrus), benzaldehyde (almond), (Z)-2-nonenal (fatty), phenylacetaldehyde (honey-like), (E, E)-2,4-nonadienal (deep fat fried), and (E,E)-2,4-decadienal (deep fat fried) were detected in all CSBs. Hexanal was the most abundant aldehyde and was more abundant in alkali-added CSB than in alkali-free CSB from most regions, with the highest concentration in CSB from WH, at 26.78 mg/kg. This was essentially consistent with the study by Xi et al. [10], which suggested that 0.1% Na_2_CO_3_ addition could significantly increase the concentration of hexanal in sourdough CSB. In addition, ethyl acetate (fruity) and 2-pentylfuran (fruity) were found in all the samples, and these volatiles also imparted CSB typical aromas [33].

PCA was performed to analyze the effect of adding alkali on the aroma compounds in CSB (Figure 5A). The first two principal components explained 53.0% of the total variance. All CSB samples were clearly separated from each other, indicating the diversity of aroma characteristics of CSBs fermented using CTSs from different regions. The score plots distinguished between alkali-added CSBs and alkali-free CSBs. Most of the alcohols, aldehydes, acids, ketones, esters, and furan were located in the positive PC1 region, and almost all acids were positioned in the positive PC2 region. Thus, the positive axis of PC2 mainly represented high content of acids, while the positive axis of PC1 represented high content of alcohols, aldehydes, ketones, esters, and furan. Compared with the corresponding alkali-free CSBs, the alkali-added CSBs were evidently shifted in the negative direction on the PC2 axis. The results showed that the effect of adding alkali on traditional CSB was mainly focused on decreasing the content of aromatic acids.

### 3.5. Correlation Analysis between Aroma Compounds and Bacteria at Genus Level

To illustrate the relationships between aroma profiles of CSBs and bacteria in the corresponding fermented dough, correlation analysis was performed on the content of aroma compounds in CSBs and the relative abundance of three dominant genera including *Lactobacillus* (*Lactobacillus* represents 25 genera, including *Companilactobacillus*, *Fructilactobacillus*, *Levilactobacillus*, *Latilactobacillus,* etc.) [34], *Pediococcus*, and *Weissella* in fermented dough, and the results are shown in Figure 5B.

The aroma compounds of CSBs were divided into two clusters using hierarchical clustering. The first cluster included most of the aldehydes (hexanal, heptanal, octanal, (*Z*)-2-heptenal, nonanal, and (*E*, *E*)-2,4-nonadienal) and ketones (acetone, 2-heptanone, 2-octanone, and 3-octen-2-one), and four kinds of alcohols (1-pentanol, 1-hexanol, 1-heptanol, and 1-nonanol). Moreover, almost all of the short-chain aldehydes (C6–C9) were distributed in this cluster, and they are generally believed to be formed by auto-oxidation of unsaturated fatty acids from wheat flour, with hexanal mainly generated from C18:2, and the others from C18:1 [35]. Hexanal is a typical product in the oxidation of linoleic acid and arachidonic acid and is considered to be negatively correlated with the bread flavor [36]. Maire et al. [35] demonstrated that heptanal, octanal, and nonanal were formed in sponge cake through a common pathway in auto-oxidation of unsaturated oleic acid. Furthermore, aldehydes with oxidative flavors in wheat bread are denoted as off-flavors when present in high concentrations [37]. *Pediococcus* and *Weissella* in fermented dough were positively correlated with short-chain aldehydes, while *Lactobacillus* was negatively correlated with them, suggesting that dough fermented by *Pediococcus* or *Weissella* had more lipid-oxidation products than the dough fermented by *Lactobacillus*. Similar results were found by Xu et al. [38], who showed that *P. pentosaceus* participated in the lipid oxidation of *broccoli* juice fermentation, affected the content of lipid degradation products including volatile aldehydes and alcohols, and enhanced the sensory properties of broccoli juice. Lipid oxidation in wheat flour and microbial metabolism are the main pathways for the formation of alcohols in fermented dough. In the lipid oxidation pathway, the alcohols are generated by lipoxidase-catalyzed oxidation of fatty acids and conversion of lipid-oxidizing compounds such as aldehydes to the corresponding alcohols [39]. *Pediococcus* and *Weissella* in fermented dough were positively correlated with 1-hexanol (green), 1-nonanol (citrus), and 1-heptanol (green), but *Lactobacillus* was negatively correlated with them. It was indicated that the fermented doughs containing *Pediococcus* and *Weissella* showed stronger lipid oxidation reactions, which provided more precursors for alcohol conversion and formed more alcohol flavor compounds than *Lactobacillus* in dough. Similarly, ketones, which are also characteristic flavor compounds in wheat bread, were principally produced via lipid oxidation [40].

Almost all of the acids, including acetic acid (sour), propanoic acid (rancid), butanoic acid, pentanoic acid, hexanoic acid, heptanoic acid, and octanoic acid (cheese), were classified in the second cluster. *Lactobacillus* in fermented dough was positively correlated with these acids in CSB, while *Pediococcus* and *Weissella* were negatively correlated with them. This can be explained in two ways: First, in the dough ecosystem, *Lactobacillus* had higher acid-producing capacity and better tolerance to acid stress than *Pediococcus* or *Weissella* [41]. Secondly, *Lactobacillus* might promote the formation of organic acids (mainly C3–C6 acids), in dough co-fermented by LAB and yeast [31].

### 3.6. Analysis of Aroma Compounds in CSB Prepared with Synthetic Microbial Community Starter

To further analyze the effects of different dominant LAB and alkali addition on the aroma profiles of CSB, referring to the microbial community analysis described above, *P. pentosaceus*, *C. crustorum*, and *W. cibaria* were combined with *S. cerevisiae* as synthetic microbial community starters, respectively. The contents of aroma compounds in CSBs are shown in Appendix A and were subjected to heatmap analysis (Figure 6). Hierarchical clustering of CSBs prepared with synthetic microbial community starters showed that alkali-added CSBs and alkali-free CSBs fermented by the same starters were clustered together, indicating that adding alkali retained the original aroma characteristics of alkali-free CSBs.

Notably, >50% of the aroma compounds in CSB fermented by *P. pentosaceus* were more abundant than those in CSB fermented by *C. crustorum* or *W. cibaria*, such as phenylethyl alcohol, 3-methyl-1-butanol (balsamic), and ethanol. Phenylethyl alcohol and 3-methyl-1-butanol are typical higher alcohols in fermented dough. Higher alcohols are converted by the corresponding branched-chain amino acids via Ehrlich pathway in an anaerobic environment; for example, phenylethyl alcohol is generated from phenylalanine [42]. Ethanol is a marker of the Embden–Meyerhof–Parnas (EMP) pathway in *S. cerevisiae* metabolism under anaerobic conditions [43]. We found that CSB fermented by *P. pentosaceus* had a strong aroma with high concentrations of Ehrlich volatiles and ethanol, which indicated that *P. pentosaceus* might be better at promoting Ehrlich and EMP pathways in *S. cerevisiae* metabolism than *C. crustorum* and *W. cibaria*, in the dough ecosystem. With the addition of alkali, the content of 17 compounds such as pentanoic acid, hexanoic acid, heptanoic acid, hexanal, octanal, 1-hexanol, and 1-octen-3-ol significantly decreased in CSB, suggesting that adding alkali could weaken the aroma profile of CSB fermented by *P. pentosaceus*.

Similarly to the results of aroma compound analysis of CSB fermented by CTS, the addition of alkali decreased almost all of the volatile acids in CSB fermented by synthetic microbial community starter. Pentanoic acid, hexanoic acid, and heptanoic acid showed significant decreases in CSBs fermented by *W. cibaria* and *P. pentosaceus*, and only pentanoic acid displayed a significant decrease in CSB fermented by *C. crustorum*, which might be related to the higher tolerance of *C. crustorum* to alkali than *W. cibaria* and *P. pentosaceus* in the dough ecosystem. Acetic acid has been demonstrated to impact the aroma of wheat flour bread negatively at high concentrations [31]. The concentration of acetic acid in CSB fermented by *W. cibaria* was the highest (average 15.92 mg/kg), >1.5-fold that in *C. crustorum*-fermented CSB (average 9.45 mg/kg). The heterofermentative fermentation property of *W. cibaria* might be responsible for the accumulation of acetic acid in the fermented dough [44].Heterofermentative *W. cibaria* directly converts glucose to lactic acid and acetic acid via the pyruvate metabolism pathway, but lactic acid is the only byproduct of glucose in homofermentative *C. crustorum*, in the dough ecosystem. Heterofermentation can also produce esters that impart a pleasant and fruity odor in bread [45]. Ethyl acetate and methyl acetate (fragrant) in high concentrations were detected in the alkali-added CSB fermented by *W. cibaria*. Notably, adding alkali significantly decreased (by >2-fold) acetic acid in *W. cibaria*-fermented CSB, and slightly decreased that in CSB fermented by *P. pentosaceus* or *C. crustorum*. Thus, it was indicated that heterofermentative *W. cibaria* accumulated more acetic acid than *P. pentosaceus* or *C. crustorum* in fermented dough, and adding alkali not only neutralized acetic acid but also promoted the formation of ethyl acetate and methyl acetate in CSB.

## 4. Conclusions

To our knowledge, this is the first systematic study to investigate the effects of adding alkali on the microbial community in CTSs and the aroma profiles of CSBs, revealing the internal relationship between LAB and aroma characteristics of CSBs. The dominant bacteria at the genus level in CTS were *Pediococcus*, *Companilactobacillus*, and *Weissella*. Adding alkali did not change the types of dominant LAB derived from CTS but changed their relative abundance in fermented dough. Adding alkali decreased the concentration of volatile acids in CSB. *Pediococcus* and *Weissella* were positively correlated with lipid oxidation flavor-related compounds in CSBs; *Lactobacillus* was positively correlated with sourness volatiles, suggesting that *Lactobacillus* had strong acid-producing capacity. The synthetic microbial community experiment exhibited that CSB fermented by *P. pentosaceus* had a strong aroma, and adding alkali weakened the flavor. CSB fermented by heterofermentative *W. cibaria* contained abundant acetic acid, and adding alkali decreased its concentration, promoted the formation of esters, and endowed CSB with a pleasant fruity aroma. Overall, the addition of alkali not only neutralizes organic acid in fermented dough, but also affects microbial metabolism, thereby reducing sourness and changing the lipid oxidation flavor and ester aroma of CSB. This study illustrates the mechanism underlying the improvement in the flavor of CSB via alkali addition and will provide a theoretical foundation for quality improvement and industrial production of steamed bread.

## Figures and Tables

**Figure 1 foods-12-00617-f001:**
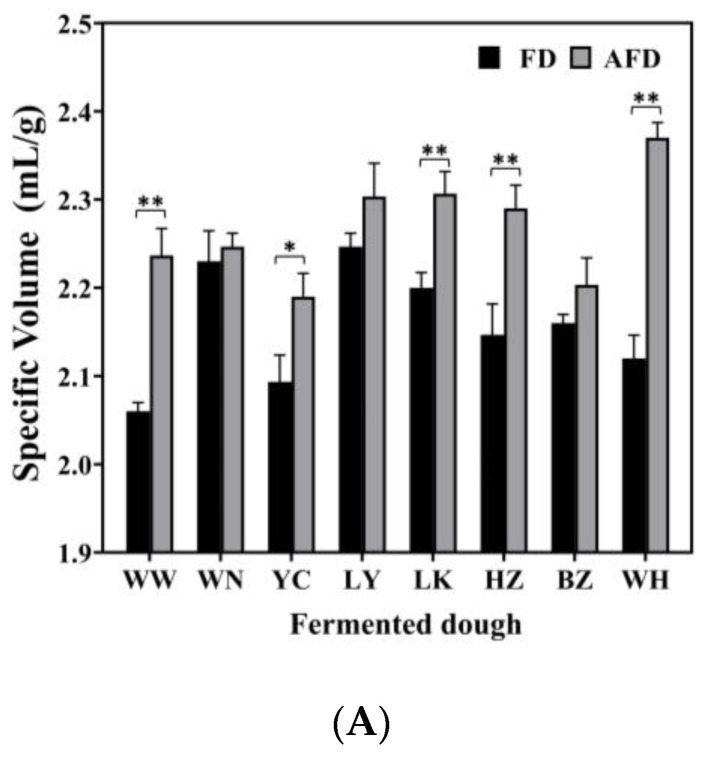
(**A**) The specific volume of alkali-free fermented wheat dough (FD) and alkali-added fermented wheat dough (AFD). (**B**) The pH values of FD and AFD. (**C**) The total titratable acid (TTA) values of FD and AFD. WW, WN, YC, LY, LK, HZ, BZ, and WH represent the FDs and AFDs fermented by Chinese traditional starters (CTSs) from eight regions of China. * *p* < 0.05; ** *p* < 0.01.

**Figure 2 foods-12-00617-f002:**
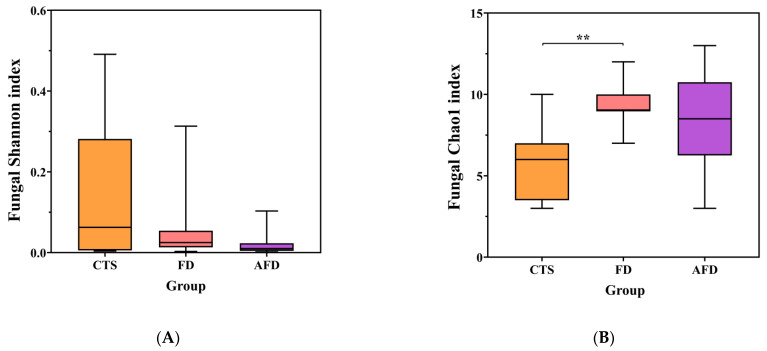
Microbial diversity indexes, including the Shannon and Chao1 indexes of CTS, FD, and AFD. (**A**) Chao1 index of fungi. (**B**) Shannon index of fungi. (**C**) Chao1 index of bacteria. (**D**) Shannon index of bacteria. * *p* < 0.05; ** *p* < 0.01.

**Figure 3 foods-12-00617-f003:**
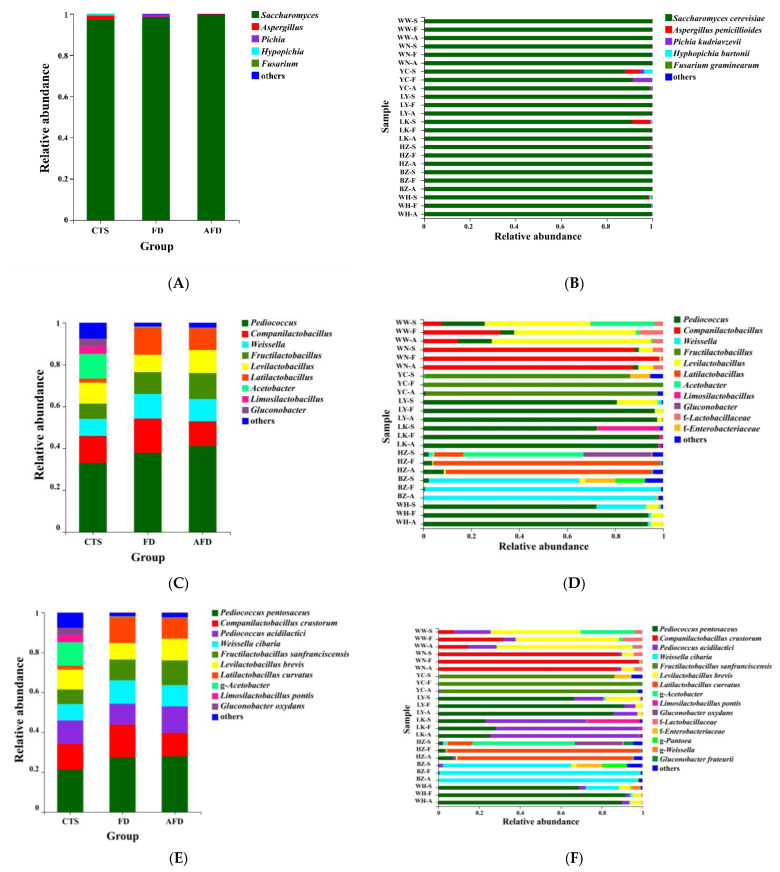
Relative abundance of microbial communities in CTS, FD, and AFD. (**A**) Fungal communities in three groups (CTSs, FDs, and AFDs from eight regions of China) at the genus level. (**B**) Fungal communities in 24 samples (CTSs, FDs, and AFDs from eight regions of China) at the species level. (**C**) Bacterial communities in three groups at the genus level. (**D**) Bacterial communities in 24 samples at the genus level. (**E**) Bacterial communities in three groups at the species level. (**F**) Bacterial communities in 24 samples at the species level. Samples are indicated using the following codes: CTS samples (WW-S, WN-S, YC-S, LY-S, LK-S, HZ-S, BZ-S, WH-S); FD samples (WW-F, WN-F, YC-F, LY-F, LK-F, HZ-F, BZ-F, WH-F); and AFD samples (WW-A, WN-A, YC-A, LY-A, LK-A, HZ-A, BZ-A, WH-A).

**Figure 4 foods-12-00617-f004:**
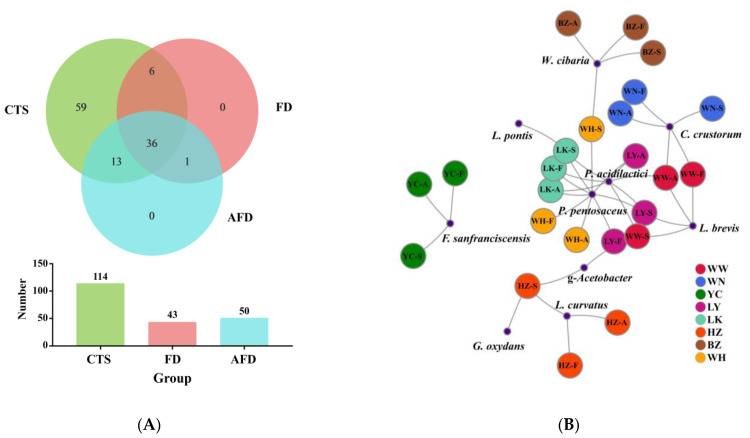
(**A**) Venn diagram displaying the distribution of shared orthologous bacterial species among CTS, FD, and AFD groups. (**B**) Co-occurrence network analysis of the bacterial communities among 24 samples at the species level.

**Figure 5 foods-12-00617-f005:**
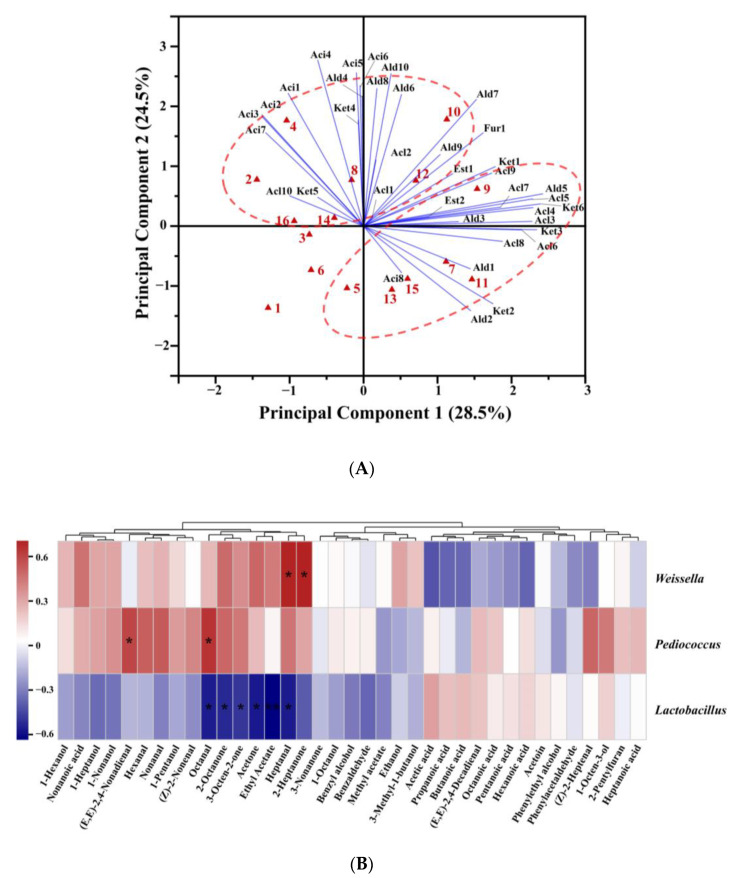
(**A**) Biplot drawn based on the first two principal components extracted by principal component analysis (PCA) of volatile compounds in alkali-free Chinese steamed breads (CSBs) and alkali-added CSBs. The names of the CSB samples and compounds in the biplot refer to Appendix A. The numbers 1–16 represent alkali-free CSBs and alkali-added CSBs fermented by CTSs from eight regions of China in the sequence WW-AB, WW-FB, WN-AB, WN-FB, YC-AB, YC-FB, LY-AB, LY-FB, LK-AB, LK-FB, HZ-AB, HZ-FB, BZ-AB, BZ-FB, WH-AB, and WH-FB. Alc, Ald, Aci, Est, Ket, and Fur represent alcohols, aldehydes, acids, esters, ketones, and furan, respectively. (**B**) Comparison of volatile compounds in *Weissella*-, *Lactobacillus*-, and *Pediococcus*-associated CSBs. A heatmap was constructed using the relative abundance of bacteria in fermented doughs at the genus level. Spearman correlation with hierarchical clustering among the volatile compounds was applied. * *p* < 0.05; ** *p* < 0.01.

**Figure 6 foods-12-00617-f006:**
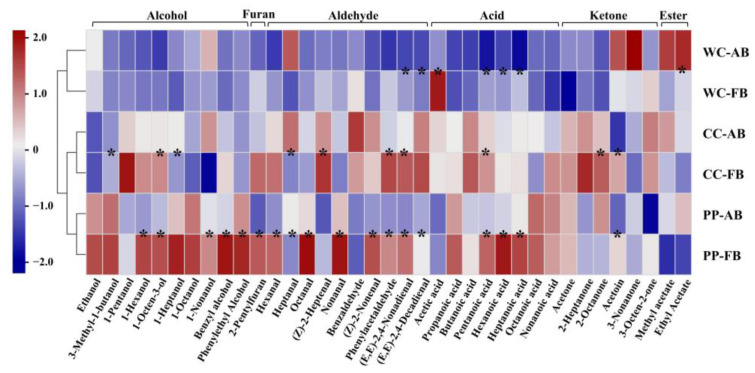
Comparison of volatile compounds in alkali-added and alkali-free CSBs fermented using synthetic microbial community starters. Pearson correlation with hierarchical clustering among the CSB samples was applied. WC-FB, CC-FB, and PP-FB indicate alkali-free CSBs fermented by synthetic microbial community starters containing *Weissella cibaria*, *Companilactobacillus crustorum*, and *Pediococcus pentosaceus*, respectively. WC-AB, CC-AB, and PP-AB symbolize alkali-added CSBs fermented by synthetic microbial community starters containing *W. cibaria, C. crustorum,* and *P. pentosaceus*, respectively. * *p* < 0.05.

**Table 1 foods-12-00617-t001:** Characteristics of CTSs sampled from different regions in China.

CTS	Sampling Site	Location	Raw Materials	Microbial Materials	Season
WW	Wuwei City, Gansu Province	37°33′ N, 102°18′ E	Wheat flour	Sourdough	Summer
WN	Weinan City, Shaanxi Province	37°30′ N, 109°36′ E	Wheat flour, Corn flour	Sourdough	Summer
YC	Yuncheng City, Shanxi Province	34°50′ N, 111°13′ E	Wheat flour, Corn flour	Sourdough	Summer
LY	Luoyang City, Henan Province	34°31′ N, 112°03′ E	Wheat flour, Corn flour	*Cucumis melo*, Rice wine	Summer
LK	Lankao City, Henan Province	34°41′ N, 114°48′ E	Wheat flour, Corn flour	*Cucumis melo*, Rice wine	Summer
HZ	Heze City, Shandong Province	35°14′ N, 115°28′ E	Wheat flour, Corn flour	*Cucumis melo*, Chinese jute (*Abutilon theophrasti*)	Summer
BZ	Binzhou City, Shandong Province	37°22′ N, 118°02′ E	Wheat flour	Sourdough	Summer
WH	Weihai City, Shandong Province	37°52′ N, 122°12′ E	Corn flour, Proso millet (*Panicum miliaceum*)	Sourdough	Summer

## Data Availability

The data presented in this study are available in Appendix A.

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
