# Peer review of "Effect of Alkali on the Microbial Community and Aroma Profile of Chinese Steamed Bread Prepared with Chinese Traditional Starter"

_foods, 2023, doi:10.3390/foods12030617_

Round 1

Reviewer 1 Report

Reviewer report for paper entitled "Effect of alkali on the microbial community and aroma profile of Chinese steamed bread prepared with Chinese traditional starter". This article represents an interesting study on the influence of alklai on the microbial diversity of starter culture and aroma production during traditional Chinese steamed bread manufacturing. The study is of interest for both scientific and industrial society to improve characteristics and functional properties of bread production. However, some points need to be improved as follows:

- In materials and methods part. Need to provide more details about the location of sampling cities in terms of GPS and the season of sampling. (lines 98-101). 

- Under section (2.4. Manufacturing of CSB using synthetic microbial community starter), I recommend to use defined or determined instead of synthetic. Need also to provide details of dough making method (line 134-136). 

- The results part is well written and data connections and interpretations are clear and provide a very good sequence for this type of research. 

- Need to link in more depth between microbial consortium (in terms of microbial physiology) and the produced aroma compounds. 

- Conclusion part need to highlight the novelty of this work in better approach. 

- Reference part needs major revision, some papers such as (Hu et al., 2022 and Huang et al., 2022, volume and page numbers are missing)

- All microbial strains in reference part need to be in italic. 

Reviewer 2 Report

General observations

In the present manuscript, the authors evaluated the impacts of the alkali addition over the microbiological and sensorial aspects of CBS prepared with Chinese traditional starter (CTS). The high-throughput sequencing revealed a complex bacterial diversity over the starter cultures, showing that the alkali addition would favor the growth of lactic acid bacteria over other microorganisms, such as acetic acid bacteria. Further physicochemical analysis also demonstrated the aroma complexity of the final product and correlated to the dominant microorganisms found in the analysis. Finally, a synthetic starter culture was constructed in other to evaluate the relationship and influence of the dominant bacteria observed in the different CTS when co-inoculated with Saccharomyces cerevisiae. The results showed were coherent. The discussion is well-elaborated throughout the manuscript; however, there are some results and correlations performed by the authors that lack explanation. English writing needs improvement.

Please, find bellow in detail my observations and questions.

Remove (or replace) the words “kind”, “being better”, among others. Avoid subjective terms alongside the text. Please, revise the manuscript to avoid it.

Lines 50 – 52: Sentence confusing and too long. Please, rewrite it.

Line 60: Unnecessary sentence. Remove it.

Section 2.1 – Specify the composition of the CTS of each region (i.e., WH was a CTS from a specific fruit; LK was a CTS from rice wine). This discrepancy is necessary so the readers can fully understand the complexity of the microbial community and, also, aid to explain differences between AFD and FD, aroma composition, acidity, among other key-sensorial characteristics.

Section 2.3. – Specify the amplification conditions of each marker (ITS and 16S rRNA gene).

Section 2.4 – The cells were washed with Ultrapure water?

Figure 1 – Add the statistical analysis among AFD and FD from each region.

Lines 194-196 – Which ecosystems?

Figure 3E and 3F – The genus Lactobacillus was recently revised, and 23 novel genera have been proposed – and widely accepted - by Zheng et al. (2020) [https://doi.org/10.1099/ijsem.0.004107]. Please, revise the scientific names accordingly in the Figure and the manuscript.

Lines 270 – 272. This increase was observed only in the LK region. In Figure 3F the relative abundance of Pediococcus pentosaceus reduced from F group to A (alkali-added? – not specified in the captions) in WH and LY regions. Reformulate the sentence to avoid generalization of erroneous data. Why were these discrepancies noted?

Line 317 – According to the Figure 3F, the addition of alkali did not change the dominant bacterial population, it only reinforced the relative abundance of the dominant species. It is correct to inquire that the alkali condition did alter the bacterial composition, but not the dominant microorganism. The only exception was in the Heze City, Shandong (HZ), where Companilactobacillus crustorum (former L. crustorum) surpassed the Acetobacter sp. Why was this change observed? Which matrix was this CTS made from? The explanation from lines 274 – 278 can not be the only explanation to it.

Reviewer 3 Report

Comments to authors

Journal: Foods

Title: ‘Effect of alkali on the microbial community and aroma profile of Chinese steamed bread prepared with Chinese traditional starter. The research is interesting, and the manuscript looks good too but still, I have fewer minor questions as mentioned below.

Minor comments:

1.     The article needs to be revised thoroughly by English experts as it contains many syntax and grammatical errors.

2.     Fig. 3E some names are missing e.g. Lactobacillus sanfranci. etc

3.     Fig. 4B should be redesigned to make it more beautiful and clear.

4.     Reference style should be same including page number, volume and issue number. The authors should recheck the references.

Round 2

Reviewer 2 Report

After the corrections, I consider the manuscript suitable for publication, in the  current form, in the Foods Journal.